

# Invasive vegetation affects amphibian skin microbiota and body condition

Obed Hernández-Gómez[1], Allison Q. Byrne[1], Alex R. Gunderson[2], Thomas S. Jenkinson[1], Clay F. Noss[1], Andrew P. Rothstein[1], Molly C. Womack[1] and Erica B. Rosenblum[1]

[1] Department of Environmental Science, Policy, and Management, University of California, Berkeley, Berkeley, CA, USA
[2] Department of Ecology and Evolutionary Biology, Tulane University, New Orleans, LA, USA

Corresponding author
Obed Hernández-Gómez,
obedhg@berkeley.edu

## ABSTRACT

Invasive plants are major drivers of habitat modification and the scale of their impact is increasing globally as anthropogenic activities facilitate their spread. In California, an invasive plant genus of great concern is *Eucalyptus*. *Eucalyptus* leaves can alter soil chemistry and negatively affect underground macro- and microbial communities. Amphibians serve as excellent models to evaluate the effect of *Eucalyptus* invasion on ground-dwelling species as they predate on soil arthropods and incorporate soil microbes into their microbiotas. The skin microbiota is particularly important to amphibian health, suggesting that invasive plant species could ultimately affect amphibian populations. To investigate the potential for invasive vegetation to induce changes in microbial communities, we sampled microbial communities in the soil and on the skin of local amphibians. Specifically, we compared *Batrachoseps attenuatus* skin microbiomes in both *Eucalyptus globulus* (Myrtaceae) and native *Quercus agriflolia* (Fagaceae) dominated forests in the San Francisco Bay Area. We determined whether changes in microbial diversity and composition in both soil and *Batrachoseps attenuatus* skin were associated with dominant vegetation type. To evaluate animal health across vegetation types, we compared *Batrachoseps attenuatus* body condition and the presence/absence of the amphibian skin pathogen *Batrachochytrium dendrobatidis*. We found that *Eucalyptus* invasion had no measurable effect on soil microbial community diversity and a relatively small effect (compared to the effect of site identity) on community structure in the microhabitats sampled. In contrast, our results show that *Batrachoseps attenuatus* skin microbiota diversity was greater in *Quercus* dominated habitats. One amplicon sequence variant identified in the family Chlamydiaceae was observed in higher relative abundance among salamanders sampled in *Eucalyptus* dominated habitats. We also observed that *Batrachoseps attenuatus* body condition was higher in *Quercus* dominated habitats. Incidence of *Batrachochytrium dendrobatidis* across all individuals was very low (only one *Batrachochytrium dendrobatidis* positive individual). The effect on body condition demonstrates that although *Eucalyptus* may not always decrease amphibian abundance or diversity, it can potentially have cryptic negative effects. Our findings prompt further work to determine the mechanisms that lead to changes in the health and microbiome of native species post-plant invasion.

## INTRODUCTION

Anthropogenic habitat modification has dramatic direct and indirect effects on wild animal populations (*Acevedo-Whitehouse & Duffus, 2009*). Invasive plants are major drivers of habitat modification, and the scale of their impacts is increasing globally (*Pysek et al., 2012*; *Van Kleunen et al., 2015*). In the USA, approximately 5,000 alien plants have been introduced into natural ecosystems, causing significant ecological and environmental degradation (*Pimentel, Zuniga & Morrison, 2005*). Invasive plants are known to affect many patterns and processes in native communities (e.g., changes in habitat structure, productivity, pH, transpiration, etc.), which in turn can have profound impacts on native species (*Pysek et al., 2012*; *Vila et al., 2011*). While the effect of invasive vegetation can vary based on the biology of the plant and age of invasion (*Hejda, Pyšek & Jarošík, 2009*), changes in native community diversity and composition have been documented in areas that have been dominated by invasive plants (*Batten et al., 2006*; *Tererai et al., 2013*; *Litt et al., 2014*).

One way that invasive plants impact native animal populations is by altering their microbial commensals. Plant invasions could influence the microbial community structure of native fauna by changing microbial communities that hosts are exposed to, by altering host physiology, or both (*Christian, Whitaker & Clay, 2015*). Invasion may alter environmental microbial reservoirs by shifting abiotic (e.g., temperature, moisture) and biotic (e.g., species diversity) conditions that affect the presence of certain microbial species (*Batten et al., 2006*; *Coats & Rumpho, 2014*). However, the association between host microbiotas and the habitat microbial pool vary among studies. Among wild populations in relatively natural habitats, some studies have found that microbiomes vary significantly with habitat type (*Bird et al., 2018*; *Bletz et al., 2017*) whereas others have found that microbiomes are relatively conserved and coevolve with hosts (*Prado-Irwin et al., 2017*). Thus, whether changes in the local microbial community structure also affect host microbial symbionts remains an open question.

In California one of the invasive plants of greatest concern are the *Eucalyptus* sp. (*Fork et al., 2015*; *Wolf & DiTomaso, 2016*). *Eucalyptus* were introduced into the state in the 1850's as a timber species (*Butterfield, 1935*), and multiple members of this genus are now abundant and ecologically successful throughout the state (*Ritter & Yost, 2009*). *Eucalyptus* leaves can alter soil nutrient availability (e.g., organic carbon, nitrogen, $O_2$) resulting in changes in soil microbial communities (*Chen et al., 2013*; *Cortez et al., 2014*). In addition, *Eucalyptus* leaf essential oils have been observed to be toxic to soil fungi and negatively affect food palatability to soil arthropods (*Martins et al., 2013*). Changes in toxicity and palatability can impact prey availability for native fauna and subsequently may alter their microbiomes (*Antwis et al., 2014*). Resulting changes in microbiomes may have important fitness consequences especially if microbial species contribute to host physiological processes (*Redford et al., 2012*). Thus, *Eucalyptus* invasions may alter the

microbiome of native fauna by changing prey availability and/or shifting the structure of microbial reservoirs.

Amphibians serve as excellent models to evaluate host-associated microbiome changes in response to habitat changes as they predate on soil arthropods and incorporate soil microbes into their microbiotas (*Loudon et al., 2014*). The skin of amphibians is a vital organ used for respiration, osmoregulation and immunity, but it is also sensitive to environmental changes, including temperature/moisture fluctuations, pollution, and infections (*Brühl, Pieper & Weber, 2011*; *Haslam et al., 2014*). In addition, amphibian skin harbors diverse microbial communities that provide protection against lethal amphibian pathogens (*Harris et al., 2009*; *Woodhams et al., 2014*). Because the skin microbiota of amphibians recruits environmental microbes (*Walke et al., 2014*), environmental changes may result in consequential alterations to the amphibian skin community structure (*Loudon et al., 2014*; *Muletz et al., 2012*). Despite the importance of habitat quality in shaping amphibian skin microbiotas, only a handful of studies have evaluated the effect of environmental changes on these communities (*Krynak, Burke & Benard, 2015*; *Costa et al., 2016*; *Krynak, Burke & Benard, 2016*; *Hughey et al., 2017*), and, to our knowledge none have assessed the effect of invasive vegetation. The link between the skin microbiota and amphibian health suggests that environmental changes like plant species invasions may negatively affect amphibian populations.

To investigate potential changes induced by invasive vegetation on environmental and host-associated microbial communities, we sampled microhabitat soil and *Batrachoseps attenuatus* skin microbiomes in both *Eucalyptus globulus* (Myrtaceae) and native *Quercus agrifolia* (Fagaceae) dominated forests in the San Francisco Bay Area. Specifically, we determined whether changes in microbial composition, diversity and stability in both soil and *Batrachoseps attenuatus* skin were associated with *Eucalyptus* or *Quercus* dominated habitat. To evaluate animal health across *Eucalyptus* and *Quercus* dominated habitats, we also measured and compared *Batrachoseps attenuatus* body condition and the presence/absence of the amphibian chytrid fungus *Batrachochytrium dendrobatidis*, which causes the lethal amphibian disease chytridiomycosis. Our results illustrate a decline in the richness of skin associated microbiota and a decrease in salamander body condition in *Eucalyptus* forest, illustrating that plant invasions may have consequences for native species.

## MATERIALS AND METHODS

### Study system

The San Francisco Bay Area (Bay Area) provides an opportunity to test the effect of *Eucalyptus* invasions on native fauna and their microbiotas. The Bay Area is home to numerous seed-producing stands of *E. globulus*, *E. pulchella*, and *E. viminalis* among a mosaic of mixed native evergreen forests and coastal scrublands. The discrete—yet interspersed—distribution of invasive and native vegetation types in the Bay Area make it an ideal location to evaluate vegetation effects on resident host-associated microbiotas while controlling for geography. In addition, multiple ground dwelling amphibians are distributed throughout the Bay Area and may be sensitive to effects of invasive *Eucalyptus*

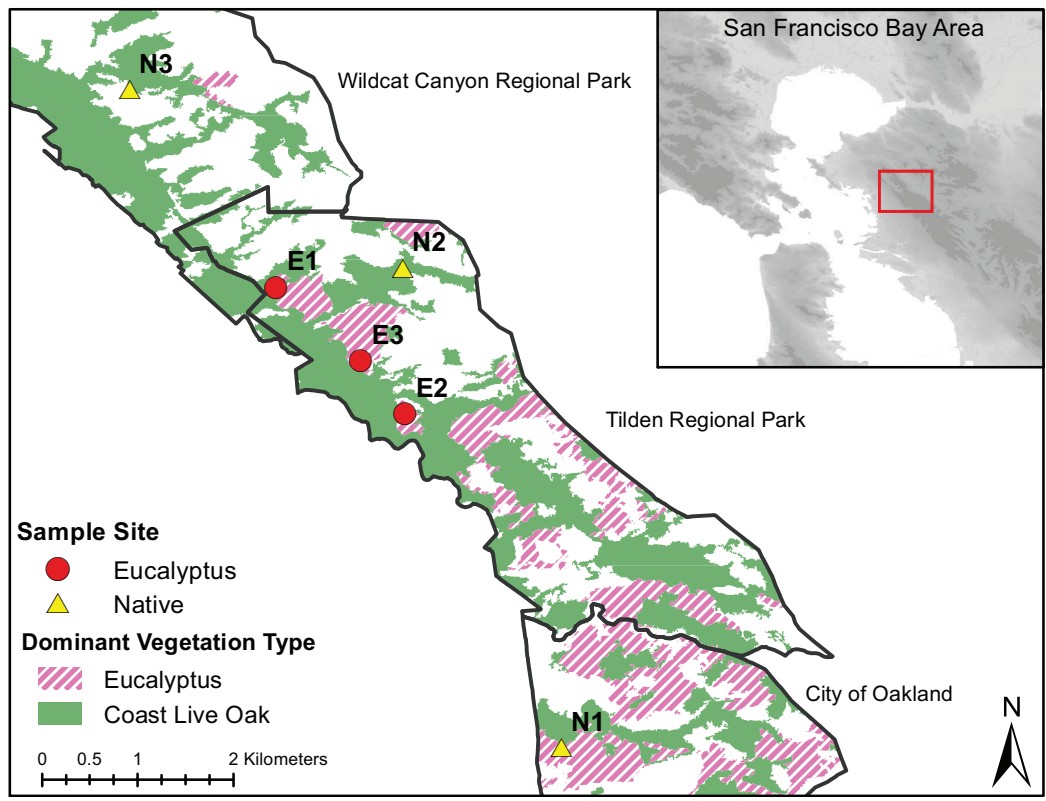

**Figure 1** **Dominant vegetation map of Wildcat Canyon Regional Park, Tilden Regional Park and UC Berkeley Campus.** Sampling sites are displayed for *Quercus* and *Eucalyptus* dominant habitats from which *Batrachoseps attenuatus* skin microbiome swabs and soil were collected.

on the skin microbiome (*Stebbins, 2003*). One amphibian species that is, present, abundant and can be easily collected throughout invasive *Eucalyptus* and native vegetation Bay Area habitats is *Batrachoseps attenuatus*. *Batrachoseps attenuatus* is an ideal focal species for testing the effects of invasive vegetation as it may be particularly sensitive to local environmental changes (e.g., *Eucalyptus* invasion) because of their highly-limited dispersal (<2 m lifetime movement; *Maiorana, 1978*) and completely terrestrial life-cycle.

We sampled *Batrachoseps attenuatus* between February 16 and March 1, 2018 within *Quercus* and *Eucalyptus* dominant forest strands in Tilden Regional Park, Wildcat Canyon Regional Park and the University of California, Berkeley campus (Fig. 1). To minimize non-vegetation influences on the skin microbiota of *Batrachoseps attenuatus*, we chose sites with similar elevation, slope and slope orientations previously described in *Sax (2002*, Table 1*)*. We excluded one of the native oak sites from *Sax (2002*; Native₃*)* as it has been eroded by a nearby stream, and instead chose a new site with similar characteristics.

## Field methodology

We handled all salamanders following a protocol approved by the University of California, Berkeley Animal Care and Use Committee (protocol # AUP-2015-01-7083-1). Access to off-campus sampling sites was granted by East Bay Regional Parks under permit # 965.

We collected salamanders by hand using gloves through log flip surveys within ~100 m of the location coordinates. New gloves were donned between the handling of each salamander. To avoid resampling individuals and ensure that salamanders were later returned to their original capture site, we marked each capture log with the individual(s) identification number. We rinsed each salamander with 250 mL of sterile water and swabbed the dorsum with a sterile cotton swab 30 times. Following microbiota sampling, we measured each salamanders' total body length and mass. All salamanders were returned to their location of capture immediately after sampling. To characterize the microhabitat microbiota, we collected ~20 mg of soil or soil swabs directly from under the logs where each salamander was captured after releasing each individual. We collected 22 *Batrachoseps attenuatus* skin swabs from *Quercus* and 28 from *Eucalyptus* dominated habitats (Fig. 1). All samples were stored on dry ice for up to 8 h and moved into a −80 °C freezer upon return to the laboratory.

## DNA extraction, amplification and sequencing

We isolated DNA from skin swab samples using the DNeasy PowerSoil DNA Isolation Kit (Qiagen N.V., Hilden, Germany) following the modifications to the manufacturer's protocol described in *Hernández-Gómez et al. (2017b)*. Soil samples were processed similarly to swabs for pre-cell lysis steps, but post-lysis steps were performed following the original manufacturer's protocol to ensure proper removal of PCR inhibitors. To control for contamination, we included unused swabs (i.e., negative extraction controls) and researcher glove swabs in our DNA extractions. We amplified the bacterial 16S rRNA V4 region using primer pair F515/R806 with the attachment of connector sequences that allows for the attachment of barcode/sequencing adaptors (*Hernández-Gómez, Hoverman & Williams, 2017a*). We ran each sample in triplicate, and each reaction consisted of 5.0 μL of template DNA, 7.5 μL of 2X MyTaq Master Mix (Bioline, Tauton, MA, USA), 1.0 μL of 1 nM forward and reverse primers, and 1.5 μL of sterile water for a total of 15 μL per reaction. PCR conditions consisted of 94 °C for 3 min, 30 cycles of 94 °C for 45 s, 50 °C for 60 s, and 72 °C for 90 s, followed by 72 °C for 10 min. We pooled amplicon triplicates and cleaned the products using the UltraClean PCR Clean-up kit (Qiagen N.V., Hilden, Germany).

We performed a second PCR on microbiota amplicons to ligate dual-index barcodes paired with Illumina sequencing adaptors (*Hernández-Gómez, Hoverman & Williams, 2017a*) to the ends of amplicons. The PCR consisted of 5.0 μL of clean amplicons, 7.5 μL 2X MyTaq Master Mix, 1.0 μL of 1 nM forward and reverse barcode primers, and 1.5 μL of water for a total of 15 μL reactions. PCR conditions consisted of 94 °C for 3 min, 5 cycles of 94 °C for 45 s, 65 °C for 60 s, and 72 °C for 90 s, followed by 72 °C for 10 min. We quantified the PCR products using a Qubit Fluorometer (Invitrogen Corp, Carlsbad, CA, USA), pooled samples in equimolar amounts, and cleaned the sample pool using the UltraClean PCR Clean-Up kit. The sample pool was submitted to the California Institute for Quantitative Biosciences Vincent J. Coates Genomics Sequencing Laboratory to be sequenced on the MiSeq plaform (Illumina Inc., San Diego, CA, USA) using the Reagent Kit V3 to produce 300 bp paired end reads.

### *Batrachochytrium dendrobatidis* detection and quantification

To detect and quantify *Batrachochytrium dendrobatidis* infection in our swabbed individuals, we performed real-time quantitative PCR (qPCR) reactions in duplicate following *Boyle et al. (2004)* with slight modifications. Each qPCR reaction consisted of 5.0 μL of 1:10 diluted template DNA, 12.5 μL of 2× TaqMan Fast Advanced Master Mix (Thermo Fisher, Waltham, MA, USA), 900 nM forward/reverse primers ITS1-3 Chytr and 5.8S Chytr (*Boyle et al., 2004*), 250 nM minor groove binder probe Chytr MGB2 (*Boyle et al., 2004*), 400 ng/μL of BSA, and 2.75 μL of molecular grade water for a total reaction volume of 25 μL. For each 96-well qPCR reaction plate, we included three replicates of a *Batrachochytrium dendrobatidis* zoospore standard dilution ranging from 100,000 to 0.1 genomic equivalents. These zoospore standards were prepared using the Bd-GPL strain CJB7–originally isolated from Kings Canyon, CA. At least three reactions per 96-well plate were designated as negative controls, with each receiving five μL of water in lieu of template DNA. To reduce the risk of laboratory contamination we set up qPCR reactions in a laminar flow hood. We ran all qPCR reactions on an Applied Biosystems StepOnePlus Real-Time PCR System (Thermo Fisher, Waltham, MA, USA), and used the manufacturer's software for standard curve analysis. We considered an average qPCR quantification of less than one genomic equivalent per swab to be negative for *Batrachochytrium dendrobatidis* infection.

## Microbiota sequence analysis

We processed raw sequencing reads using Trimmomatic (*Bolger, Lohse & Usadel, 2014*) to remove adapter sequences, bases below threshold quality of phred-20 from both ends of reads, and any resulting reads under 30 bp. We paired reads that passed initial quality control using PANDAseq (*Masella et al., 2012*). Only reads that paired successfully were employed in subsequent analysis.

Our microbiota sequence analysis consisted of established sequence read processing pipelines to filter erroneous reads, generate an amplicon sequence variant (ASV; error-corrected unique DNA sequences) table, create a representative sequence phylogeny and assign taxonomy to ASVs. We chose to use ASVs rather than operational taxonomic units (OTUs), because ASVs provide greater resolution in amplicon differentiation (*Callahan, McMurdie & Holmes, 2017*). ASV variants can be denoted by single nucleotide differences based on sequencer error correction methodologies, which surpass the accuracy obtained by OTU grouping which implements an arbitrary sequence difference threshold to cluster amplicons. We processed the resulting read file using the Quantitative Insights Into Microbial Ecology version 2.2018.4 (QIIME2) pipeline (*Bolyen et al., 2018*). We processed reads using the DADA2 plugin to quality filter, dereplicate, remove chimeras and denoise reads using default settings (*Callahan et al., 2016*). We generated a phylogeny using MAFFT aligned representative ASV sequences in FastTree2 to be used in alpha and beta analyses (*Katoh & Standley, 2013*; *Price, Dehal & Arkin, 2010*). We applied a pre-trained Naïve Bayes classifier on the Greengenes 13_8 database to assign taxonomy to each ASV at the genus level (*DeSantis et al., 2006*). We ran the ASV table through the package decontam in R to identify ASV's associated with glove

samples/negative extraction controls and removed identified contaminants from soil/swab ASV table (*Davis et al., 2018*). In addition, we filtered out any ASV's whose taxonomy matched chloroplast or mitochondria as these were not the target of our amplification protocol. To standardize sequencing depth throughout all samples, we rarefied the filtered ASV table to 5,115 sequences per sample. After filtering out 16S rRNA V4 amplicon sequence reads by base pair quality and length, we processed 6,706,635 reads using QIIME2 to produce 16,176 ASV's following contaminant and non-bacteria taxonomy removal. We deposited raw 16S rRNA V4 amplicon sequencing data into the NCBI Sequence Read Archive (project Accession Number: PRJNA574188).

We transferred the rarefied ASV table and Newick phylogeny to R (version 3.5.1) for further analyses. A Shapiro–Wilk test in R was implemented on all univariate dependent variables to evaluate normality prior to statistical model selection. We calculated three distinct alpha diversity metrics using the R packages *vegan* and *picante*: community richness (i.e., number of ASVs observed per sample), evenness (i.e., Shannon diversity indices) and phylogenetic diversity (i.e., Faith's phylogenetic diversity). We calculated these metrics in order to evaluate differences in the number of ASVs (community richness), distribution of ASV frequencies within samples (Shannon diversity index), and phylogenetic representation (Faith's phylogenetic diversity) across sample groups. To assess differences in community composition across samples, we applied the R packages *GuniFrac* and *vegan* to calculate three separate beta diversity metrics: pairwise unweighted/weighted UniFrac distances and Bray–Curtis dissimilarities. We chose to include these beta diversity metrics as they account for differences in presence/absence of phylogenetic lineages among samples (unweighted UniFrac), abundance-based differences in phylogenetic lineages among samples (weighted UniFrac), and non-phylogenetic abundance-based differences among samples (Bray–Curtis). In addition, we computed the core microbiomes (i.e., ASVs shared among 70% of individuals) of *Eucalyptus* soil, *Eucalyptus* salamander skin, *Quercus* soil and *Quercus* salamander skin samples.

## Statistical analysis

*Effects of Eucalyptus invasion on soil bacterial composition, diversity, and stability*: We evaluated differences in soil community diversity, composition and community homogeneity between *Quercus* and *Eucalyptus* dominated habitats. To evaluate differences in alpha diversity between soil communities sampled in *Quercus* and *Eucalyptus* dominated habitats, we implemented community richness, Shannon diversity indices and phylogenetic diversity as dependent variables, habitat type as a fixed variable and site identity as a random variable in linear mixed models (R package *lme4*; a negative binomial error distribution was used to evaluate richness and Gaussian to evaluate Shannon diversity/phylogenetic diversity models). The significance of the predictor variable was calculated with likelihood ratio tests (LRT). To characterize the strength and significance of soil community compositional and structural differentiation among habitat types, we implemented three PERMANOVA tests (R package *vegan*) using weighted UniFrac, unweighted UniFrac, and Bray–Curtis dissimilarity matrices as dependent variables and site identity as a random variable. We produced NMDS plots
using the three dissimilarity matrices to visualize clustering of samples by habitat and site identity using the R package *phyloseq*. To assess whether habitat type influenced variation in soil community structure, we performed three separate multivariate of homogeneity of group dispersions analyses (R package *vegan*) using the beta diversity metrics. For each beta diversity metric, we used a one-way ANOVA to test differences in point-to-centroid distances between soil samples obtained from *Eucalyptus* and *Quercus* dominated habitats. We implemented an indicator species analysis using the R package *indicspecies* to identify ASVs whose relative abundance differs between soil samples collected in *Eucalyptus* and *Quercus* dominated habitats.

*Effects of Eucalyptus invasion on Batrachoseps attenuatus bacterial composition, diversity and stability:* To assess whether salamander skin microbiota sampled in *Quercus* or *Eucalyptus* dominated habitat differ in their overlap with microhabitat microbiota, we calculated average unweighted UniFrac, weighted UniFrac, and Bray–Curtis dissimilarity values between every salamander skin sample and all environmental samples in its corresponding site. We used these dissimilarities as dependent variables in a generalized linear mixed model using a logit distribution with habitat type as a fixed variable and site as a random variable. We also evaluated whether skin microbial community diversity, composition and heterogeneity differed between *Batrachoseps attenuatus* sampled in *Quercus* and *Eucalyptus* dominated habitats. We assessed differences in community richness, Shannon diversity indices, and phylogenetic diversity between salamander skin communities sampled in *Quercus* and *Eucalyptus* dominated habitats using linear mixed models as described above. We ran PERMANOVA tests using weighted UniFrac, unweighted UniFrac, and Bray–Curtis dissimilarities as dependent variables, habitat type as a fixed variable and site identity as a random variable to evaluate variation in community composition and structure. We produced NMDS plots using the three dissimilarity matrices to visualize clustering of skin samples by habitat and site identity. We also performed three separate multivariate of homogeneity of group dispersions analyses to evaluate how habitat type influenced variation in community structure among individuals in the same group. For each beta diversity metric, we used a one-way ANOVA to test differences in point-to-centroid distances between *Eucalyptus* and *Quercus* salamander skin microbial communities. We implemented an indicator species analysis using the R package *indicspecies* to identify ASVs whose relative abundance differs between salamander skin samples collected in *Eucalyptus* and *Quercus* dominated habitats. Lastly, we assessed patterns of isolation by distance (IBD) by comparing our three dissimilarity matrices of community composition to a Euclidean geographic distance matrix between sampling locations. We tested for significance of IBD through mantel tests using R package *ade4* (*Dray & Siberchicot, 2018*).

*Associations between Eucalyptus invasion and Batrachoseps attenuatus body condition:* We derived body condition indices for each salamander by obtaining the least squares regression residuals of mass and total body length. This methodology has been widely used to accurately evaluate estimates of body reserves in amphibians and other vertebrates (*Ardia, 2005*; *Schulte-Hostedde et al., 2005*). We omitted two salamanders from the body condition analysis because they lost their tail immediately before or during sampling.
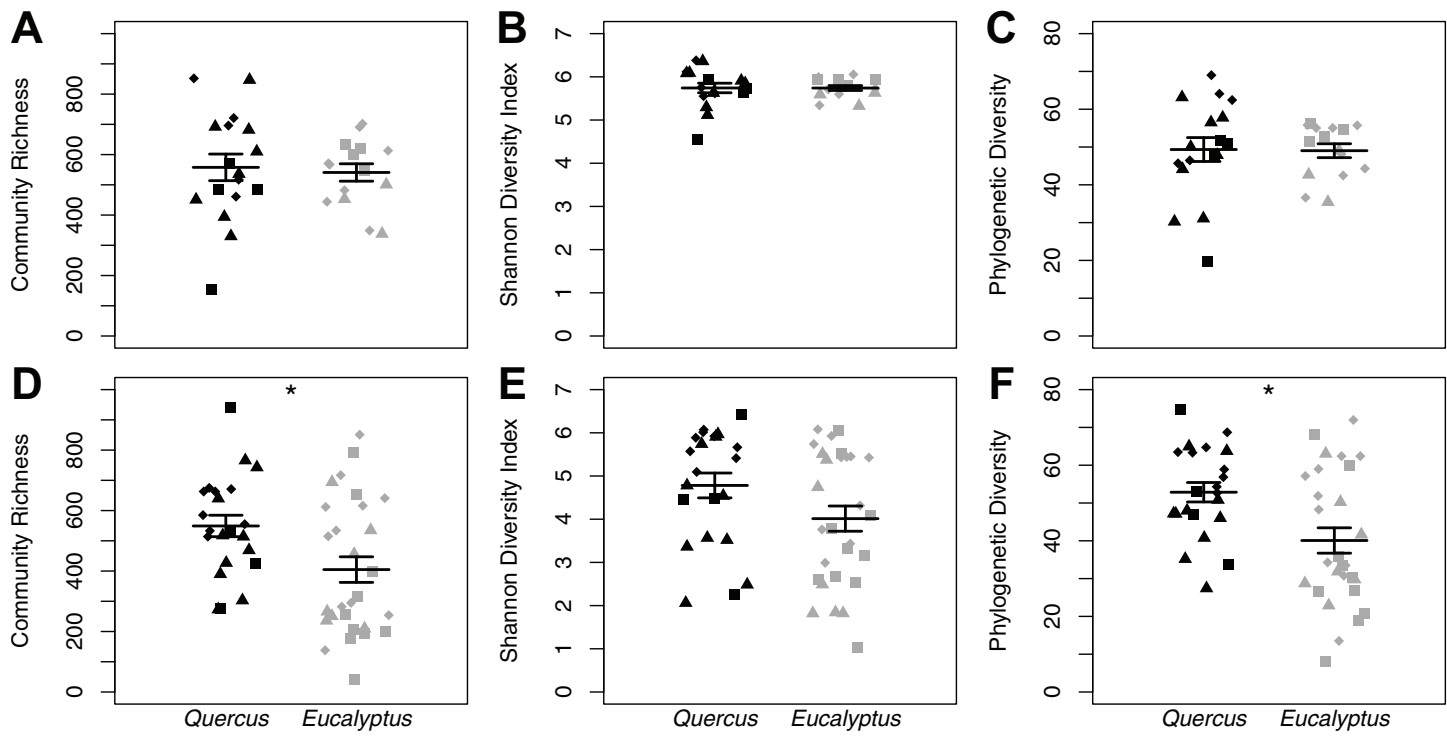

**Figure 2 Alpha diversity dot plots with mean (long horizontal line) and standard errors (short horizontal line).** Data is presented for microhabitat soil (A–C) and *Batrachoseps attenuatus* skin (D–F) microbiota samples. Alpha diversity measures presented include community richness (A and D), Shannon Diversity index (B and E), and Phylogenetic Diversity (C and F). Each point represents the bacterial skin community of an individual sample; point color indicates dominant vegetation of the habitat (Black—*Quercus agrifolia* and gray—*Eucalyptus globulus*) and shape indicates site identity (square—site 1, triangle—site 2, and diamond—site 3).

We tested differences in body condition indices between individuals sampled in *Eucalyptus* and *Quercus* dominated habitats using a linear mixed model (R package *lme4*) with site identity as a random factor. Lastly, we evaluated whether there were correlations between skin alpha diversity metrics (richness, Shannon Diversity Index, and Faith's Phylogenetic Diversity) and body condition using Pearson correlation tests.

# RESULTS

## *Eucalyptus* invasion has a small effect on microhabitat soil microbial community composition

We found no differences in soil microbial alpha diversity between *Eucalyptus* and *Quercus* dominated habitats (community richness: LRT = 0.086, $p = 0.770$; Shannon diversity index: LRT = $1 \times 10^{-4}$, $p = 0.994$; phylogenetic diversity: LRT = 0.0071, $p = 0.933$; Figs. 2A–2C). In contrast, habitat type explained a small proportion of the variation in community composition of soil samples (unweighted UniFrac: pseudo-$F_{1, 30}$ = 1.73, $R^2 = 0.05$, $p = 0.013$; weighted UniFrac: pseudo-$F_{1, 30}$ = 2.38, $R^2 = 0.07$, $p = 0.021$; Bray–Curtis: pseudo-$F_{1, 30}$ = 2.12, $R^2 = 0.07$, $p = 0.002$; Figs. 3A–3C). We did not observe significant differences in community heterogeneity among soil samples obtained from *Eucalyptus* and *Quercus* dominated habitats (unweighted UniFrac: pseudo-$F_{1, 30}$ = 3.21, $p = 0.083$; weighted UniFrac: pseudo-$F_{1, 30}$ = 2.89, $p = 0.100$; Bray–Curtis: pseudo-$F_{1, 30}$ = 1.03,

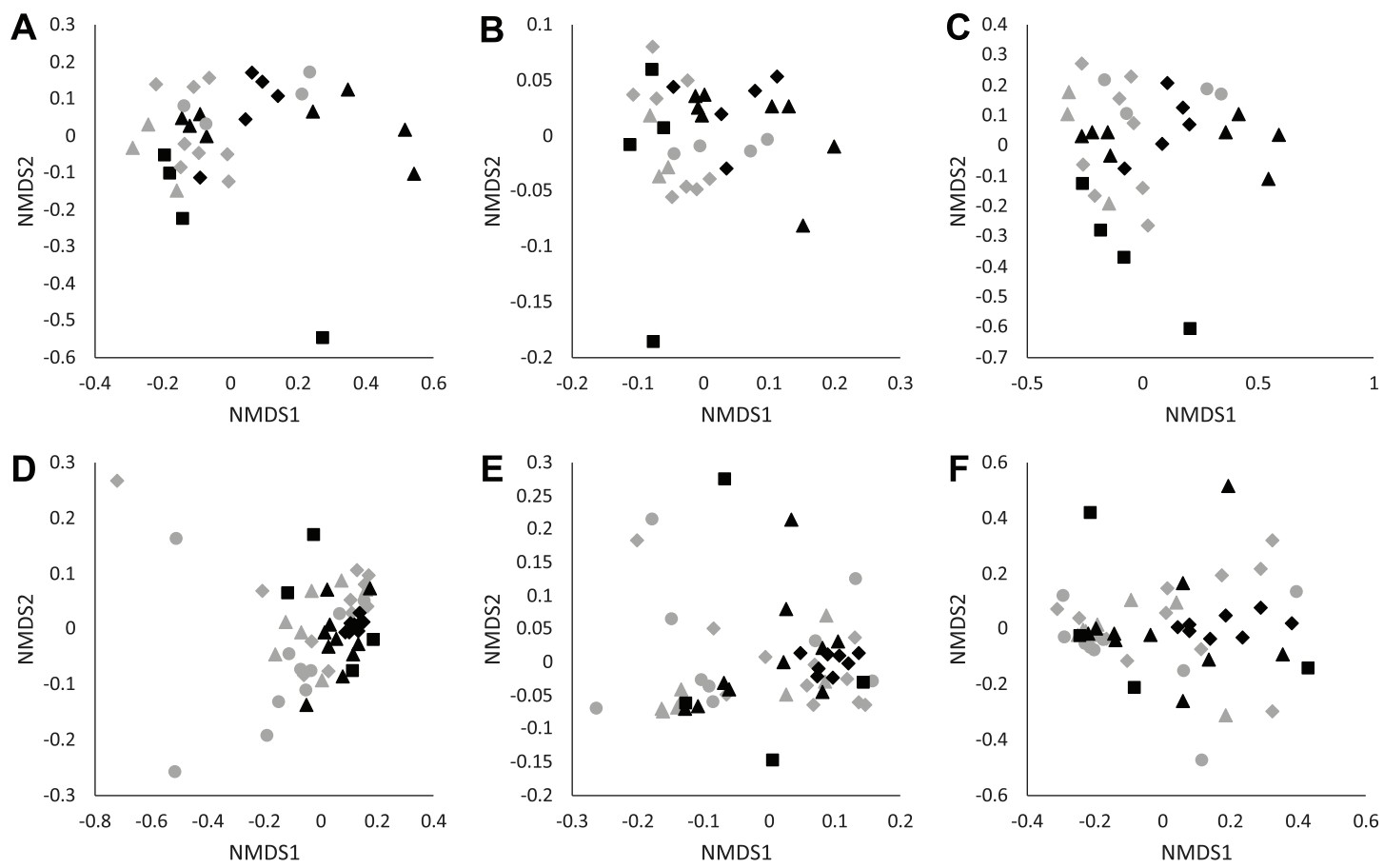

**Figure 3** NMDS plots of unweighted UniFrac (A and D), weighted UniFrac (B and E), and Bray–Curtis (C and F) distance matrices from microhabitat soils (A–C) and *Batrachoseps attenuatus* skin microbiota samples (D–F). Each point represents the bacterial skin community of an individual sample; point color indicates dominant vegetation of the habitat (Black—*Quercus agrifolia* and gray—*Eucalyptus globulus*) and shape indicates site identity (square—site 1, triangle—site 2 and diamond—site 3).

*p* = 0.319). The core microbial communities of *Eucalyptus* dominated habitat soil samples (16 ASVs) were composed of ASV's characterized as Actinobacteria (two ASVs; average of 0.38% of reads across all samples) and Proteobacteria (14 ASVs; 5.91%). In contrast, the core microbiome of *Quercus* dominated habitat soil samples was composed of eight ASV's characterized as Actinobacteria (two ASVs; 0.37%), Bacteroidetes (one ASV; 0.19%), Proteobacteria (four ASVs; 2.66%), and Verrucomicrobia (one ASV; 0.93%). The indicator species analysis identified 216 ASVs that differed significantly between habitat types with 122 associated with *Eucalytpus* soil samples and 94 associated with *Quercus* soil samples (Table S1). ASVs associated with either sample group were under a relative abundance of 1% indicating that only rare ASVs are influenced by cover type.

### *Batrachoseps attenuatus* skin microbiota diversity is greater in native *Quercus* dominated habitats

We did not observe differences in composition and structure between skin and corresponding soil samples among *Quercus* and *Eucalyptus* dominated habitats for

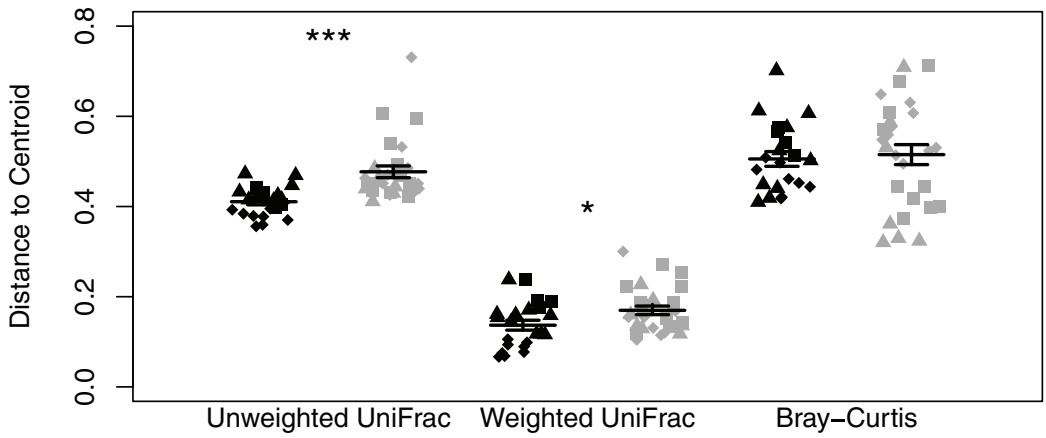

**Figure 4 Dot plot of multivariate homogeneity of groups dispersions (betadisper) of *Batrachoseps attenuatus* skin microbiota samples collected in *Quercus agrifolia* and *Eucalyptus globulus* dominant habitats.** Each point represents the bacterial skin community of an individual sample; point color indicates dominant vegetation of the habitat (Black—*Quercus agrifolia* and gray—*Eucalyptus globulus*) and shape indicates site identity (square—site 1, triangle—site 2, and diamond—site 3). Significant differences are marked with asterisks (* if $0.01 < p < 0.05$, ** if $0.001 < p < 0.01$, and *** if $p < 0.001$).

unweighted UniFrac (LTR = 0, $p = 1.000$, mean ± SD *Quercus* = 0.66 ± 0.06, mean ± SD *Eucalyptus* = 0.73 ± 0.08), weighted UniFrac (LRT = 0.673, $p = 0.412$, 0.32 ± 0.11, 0.39 ± 0.11), or Bray–Curtis dissimilarities (LRT = 0, $p = 1.000$, 0.86 ± 0.09, 0.94 ± 0.05). Microbial community richness and phylogenetic diversity were significantly lower for salamanders in *Eucalyptus* dominated habitat compared to salamanders sampled in *Quercus* habitats (community richness: LRT = 4.07, $p = 0.044$; Shannon diversity index: LRT = 1.75, $p = 0.185$; phylogenetic diversity: LRT = 4.43, $p = 0.035$; Figs. 2D–2F). Habitat type significantly explained a small portion of the differentiation in the skin composition and structure of *Batrachoseps attenuatus* (unweighted UniFrac: pseudo-$F_{1,\,48}$ = 2.48, $R^2 = 0.05$, $p = 0.001$; weighted UniFrac: pseudo-$F_{1,\,48}$ = 2.64, $R^2 = 0.05$, $p = 0.031$; Bray–Curtis: pseudo-$F_{1,\,48}$ = 2.61, $R^2 = 0.05$, $p = 0.004$; Figs. 3D–3F). We noted significantly higher heterogeneity among *Batrachoseps attenuatus* skin microbiotas from *Eucalyptus* dominated habitat than *Quercus* dominated habitat for UniFrac metrics (unweighted UniFrac: pseudo-$F_{1,\,48}$ = 4.13, $p = 0.048$; weighted UniFrac: pseudo-$F_{1,\,48}$ = 12.75, $p < 0.001$) but not for Bray–Curtis dissimilarities (pseudo-$F_{1,\,48}$ = 0.295, $p = 0.590$; Fig. 4). We found a significant association with geographic distance and Bray–Curtis dissimilarity (Mantel Bray–Curtis: $r = 0.17$, $p = 0.017$) but no patterns of IBD with both our weighted and unweighted UniFrac matrices (Mantel unweighted UniFrac: $r = -0.04$, $p = 0.581$; weighted UniFrac: $r = 0.08$, $p = 0.152$).

The core microbiota of salamander skin samples collected in *Eucalyptus* dominated habitat was comprised entirely of ASVs assigned to the phylum Proteobacteria (eight ASVs; 32.27% of reads per sample). The core microbiota of salamander skin samples collected in *Quercus* dominated habitat was richer in that it possessed 32 ASVs assigned to the phyla Actinobacteria (12 ASVs; 2.29%), Chloroflexi (1 ASV; 0.19%), Proteobacteria

(17 ASVs; 25.29%), and Verrucomicrobia (2 ASVs; 0.50%). As observed in the soil core microbial communities, salamander skin core ASVs consisted mostly of rare taxa (i.e., less than 1%). However, we found one skin ASV identified as *Bordetella petrii* to dominate skin communities in *Eucalyptus* (29.81%) and *Quercus* (20.45%) dominated habitats.

The indicator species analysis identified 294 ASVs that differed significantly between habitat types with 36 associated with *Eucalyptus* salamander skin samples and 258 associated with *Quercus* salamander skin samples (Table S2). As observed in the soil sample indicator analysis, a majority of associated ASVs were rare (relative abundance <1%). However, one ASV identified to the family Chlamydiaceae was significantly associated with and abundant in salamanders sampled from *Eucalyptus* dominated habitats (relative abundance mean ± SD: 4.32% ± 9.95%, range: 0–44.61%) compared to individuals collected in *Quercus* dominated habitats ($8.89 \times 10^{-5}$% ± $2.94 \times 10^{-4}$%, $0–1.17 \times 10^{-3}$%).

### *Batrachoseps attenuatus* body condition is higher in native *Quercus* dominated habitat

Body condition indices differed significantly between salamanders sampled in *Eucalyptus* and *Quercus* dominated habitats (LTR = 5.38, $p = 0.020$). Salamanders sampled in *Quercus* dominated habitat possessed higher body condition indices than those sampled from the *Eucalyptus* dominated habitat (Fig. 5). We did not find significant correlations between body condition indices and skin microbial community richness ($R = 0.13$, $p = 0.385$), Shannon diversity index ($R = 0.13$, $p = 0.359$), or Faith's phylogenetic diversity ($R = 0.15$, $p = 0.297$). One salamander (OHG-47_S202) tested positive for *Batrachochytrium dendrobatidis* with a low average infection load of 41.59 zoospore equivalents (Table S1). The prevalence of infection in our study sites overall was 2.0% (1/50), and prevalence did not vary significantly between our *Quercus* and *Eucalyptus* dominated sites ($t = -1$, df = 20, $p = 0.329$).

## DISCUSSION

We observed no differences in richness, little variation in community composition and structure, and similar differentiation between soil and skin samples across *Eucalyptus* and *Quercus* dominated sites. Thus, the filter acting on microbial richness is likely operating at the host level, rather than the passive uptake of different soil microbial communities.

At the individual level, we found differences in native salamander skin microbial diversity and body condition associated with *Eucalyptus* invasion. We also observed higher salamander skin microbial composition heterogeneity and relative abundances of an ASV identified to the family Chlamydiaceae in salamanders inhabiting *Eucalyptus* dominated habitats.

### Effects of invasion on the local environment's microbial community

We found no differences in soil microbiome diversity between native and *Eucalyptus* sites. In addition, a small fraction of the variation in community composition among soil samples was explained by vegetation type. This finding is surprising given the documented shifts in soil microbiomes following plant invasions (*Zhang et al., 2018*) and the unique

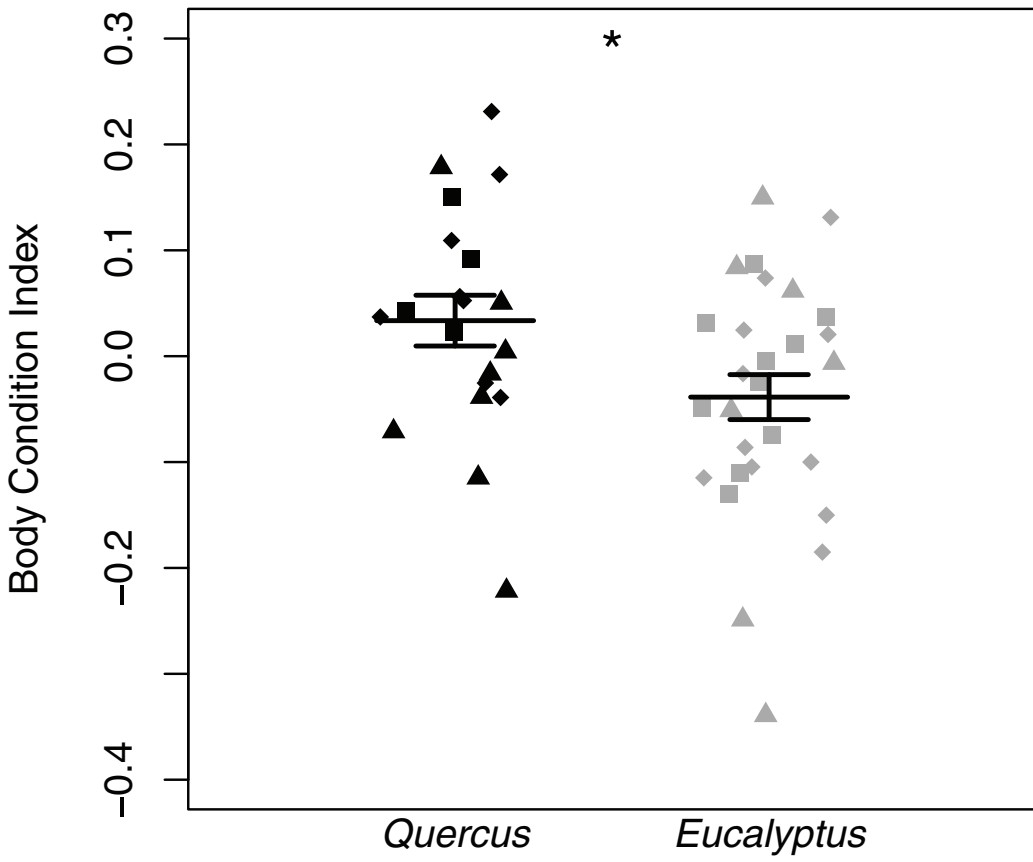

**Figure 5** *Batrachoseps attenuatus* **body condition index dot plots with mean (long horizontal line) and standard errors (short horizontal line).** Each point represents the bacterial skin community of an individual sample; point color indicates dominant vegetation of the habitat (Black—*Quercus agrifolia* and gray—*Eucalyptus globulus*) and shape indicates site identity (square—site 1, triangle—site 2, and diamond—site 3). Significant differences are marked with asterisks (* if $0.01 < p < 0.05$, ** if $0.001 < p < 0.01$, and *** if $p < 0.001$).

chemical properties of *Eucalyptus* leaf litter (*Martins et al., 2013*). The small differences observed could be driven by an inability to capture the entire microbial community with 16S sequencing. For example, we did not measure fungal diversity, a portion of the soil microbiome that has been shown to vary by space and dominant vegetation (*Sterkenburg et al., 2015*). In addition, lack of variation in soil microbiota among habitats could be driven by conservation of microbial communities in microhabitats. For example, our soil samples were all collected under logs (where the salamanders were found), which could buffer the soil from an accumulation of *Eucalyptus* leaf litter, desiccation and bacteria dispersal from forest floor microbiota (*Mäkipää et al., 2017*).

### Differences in skin microbial alpha diversity of *Batrachoseps attenuatus* associated with *Eucalyptus* invasion

We found no difference between the two habitats in the overlap between salamander skin and the local soil. In both habitats, the dissimilarities between the skin microbial

communities and those in the environment reservoirs are similar. These observations suggest that the filter acting on microbial richness is most likely operating at the host level rather than at the host-environment interface.

Microbial richness was higher in skin swab samples collected from salamanders in native *Quercus* habitat than in those collected in *Eucalyptus* habitat. This difference might be driven by the dropout of rare taxa–or the proliferation of already abundant members of the microbiota (e.g., Chlamydiaceae)–in the *Eucalyptus* samples. While we cannot differentiate among these two possibilities using 16S rRNA amplicon sequencing alone, this pattern could reflect microbial community changes due to environmental filters present in the *Eucalyptus* dominated habitat. Changes in the relative abundance of core bacteria and alpha diversity have been documented in previous studies assessing the effects of environmental changes (e.g., captivity and changes in environmental reservoirs) on the skin microbiota of amphibians (*Becker et al., 2014*; *Wuerthner, Hernández-Gómez & Hua, 2019*). Soils in *Eucalyptus* dominated plots can be a harsh environment for certain plants and microbes due to allelopathic chemicals leeched via leaves and roots, increase of soil water repellency, and changes to soil chemistry (*Behera & Sahani, 2003*; *Dellacassa et al., 1989*; *Ruwanza et al., 2015*). This harsh soil environment could exclude some taxa that are rare in the soil, but abundant on the skin of amphibians (*Walke et al., 2014*). Although we did not detect differences in soil microbial alpha diversity between *Quercus* and *Eucalyptus* soils, a negative effect of *Eucalyptus* on salamander-associated bacteria is still possible. While a direct link between amphibian skin microbial community diversity and skin health (e.g., resistance to the pathogen *Batrachochytrium dendrobatidis*) has not been established (*Jimenez & Sommer, 2016*), richer communities can possess greater functional diversity that may enhance the protective function of the skin microbiome (*Hernández-Gómez, Briggler & Williams, 2019*). As such, we recommend that future studies characterize the functionality of amphibian skin communities in the context of plant invasions to determine whether relevant microbial functions are affected as well.

## Differences in community heterogeneity and relative abundance of Chlamydiaceae in *Batrachoseps attenuatus* skin microbiota associated with *Eucalyptus* invasion

We did not observe clear differences in microbial community composition on the skin of salamanders found in native oak compared to *Eucalyptus* habitat. Similar microbial composition across habitats could be driven by a number of factors related to the assembly of these communities such as: geographic distance among sites, microhabitat environment characteristics, and strong associations between *Batrachoseps attenuatus* and members of the core microbiota. For geographic distance, we found that Bray–Curtis dissimilarity matrices were significantly correlated with distance between sites, indicating that patterns of IBD may be an important driver of observed variation and may supersede large-scale environmental filters on abundant bacteria (e.g., vegetation type, elevation, slope). Additionally, *Batrachoseps attenuatus*' high site fidelity, relatively short dispersal distances and association with fallen logs may decrease differences compared to

surrounding leaf litter, which may show greater microbial composition differences between sites (*Welsh & Droege, 2001*). This result is in contrast to similar studies of *Salamandra salamandra*, where individuals residing in different habitats (ponds vs. streams) possess distinct skin microbial communities (*Sanchez et al., 2017*). Finally, given the potential of microbiomes to influence individual fitness, salamanders' microbes could be under selection for retaining certain microbial profiles regardless of environment. For example, *Kueneman et al. (2014)* identified species specific skin microbiotas among co-habiting amphibians across distinct habitats, suggesting a strong association between amphibian species and certain microbial skin symbionts. Therefore, there may be a skin microbiome profile for *Batrachoseps* sp. that is consistent accross all potential habitats in these salamanders' broad geographic distribution.

Despite overlap in community composition among *Batrachoseps attenuatus* in *Eucalyptus* and *Quercus* dominated habitats, we noted greater phylogenetic heterogeneity in the skin microbiota of *Eucalyptus* inhabiting salamanders than in that of native *Quercus* residents. While we controlled for the effect of geography, altitude and slope among our sites, it is possible that differences in microhabitat abiotic (e.g., soil chemistry) and biotic factors (e.g., understory vegetation, forest stand age) may have disproportionate effects on the persistence of distinct rare bacteria in these salamanders. The loss of bacterial species in salamanders inhabiting *Eucalyptus* habitat may vary by individual- and site-specific factors that were not measured in the current study. Heterogeneous responses of microbial communities have been demonstrated for soil microbes in response to silviculture and agricultural practices (*Degrune et al., 2017*). Similarly for host-associated microbiotas, variability in beta diversity has been observed for microbial symbiont communities in response to environmental stressors (e.g., increased temperature, acidification, pollution; *Zaneveld, McMinds & Thurber, 2017*).

Despite an absence of differentiation in skin microbial community composition and structure between salamanders sampled in *Eucalyptus* and *Quercus* dominant habitats, we did observe a high relative abundance and almost exclusive presence of an ASV identified to the family Chlamydiaceae in *Eucalyptus* salamander skin. This single ASV difference is significant, as members of the family Chlamydiaceae have been identified as potential salamander pathogens (*Martel et al., 2012*). Systematic infection with Chlamydiaceae pathogens in salamanders has been documented to result in anorexia, lethargy, edema, abnormal gate, and death (*Martel et al., 2012*). Although we did not observe any signs of disease in the *Batrachoseps attenuatus* sampled in *Eucalyptus* dominated habitats (other than lower body condition indices), it is possible that an increase in Chlamydiaceae in the skin of salamanders results from microbial dysbiosis (*Prado-Irwin et al., 2017*). Other skin microbiota studies on terrestrial salamanders in the San Francisco Bay Area have identified Chlamydiaceae taxa on the skin microbial communities; however, the relative abundances of this taxa are usually rare as we observed in salamanders inhabiting *Quercus* habitat (*Bird et al., 2018*). While we cannot effectively link the increase in Chlamydiaceae to infectious disease in our salamanders, it is important that future studies on terrestrial salamanders evaluate

whether this bacterial group poses a threat to the health of amphibians in disturbed habitats.

### Potential effects of *Eucalyptus* invasion on *Batrachoseps attenuatus* body condition

We found that the body condition index of salamanders in *Eucalyptus* forest was significantly lower than those found in *Quercus* woodlands. The effect on body condition is consistent with prior work demonstrating a negative effect of introduced *Eucalyptus* on amphibian diversity (*Fork et al., 2015*; *Russell & Downs, 2012*). Importantly, our results also suggest that although *Eucalyptus* may not always decrease amphibian abundance or diversity (*Keane & Morrison, 1990*; *Sax, 2002*), it can have more cryptic negative effects. Lower body condition means that *Batrachoseps attenuatus* in *Eucalyptus* forest may have less energy reserves (and potentially lower fitness) than those in native *Quercus* woodland (*Schulte-Hostedde et al., 2005*). In our study, body condition was not correlated with specific skin microbiome characteristics. Although we cannot determine what proximate factor is driving body condition decline in *Eucalyptus* habitat, there are multiple possible explanations including decreased prey availability. For example, *Fork et al. (2015)* found lower arthropod richness and lower abundance of some arthropod classes in *Eucalyptus* relative to oak woodland habitat—although an earlier study did not find differences in leaf litter arthropod richness in *Eucalyptus* vs. *Quercus* woodlands; *Sax, 2002*. In addition, *Eucalyptus* leaf extract has been found to compromise chemical communication in salamanders, and this could contribute to stress in *Batrachoseps* residing in invasive vegetation dominated habitat (*Iglesias-Carrasco et al., 2017*). Ultimately, a more comprehensive comparison of abiotic and biotic characteristics in relevant microhabitats is needed to fully understand the effects of invasive vegetation on terrestrial salamanders.

## CONCLUSIONS

To our knowledge, our study is the first to demonstrate changes in the microbial communities of native hosts associated with plant invasion (*Eucalyptus*). Interestingly, we found differences in the skin microbial community of the native salamander *Batrachoseps attenuatus*, but no differences in soil microbial communities between *Eucalyptus* and *Quercus* habitats. We also found decreased body condition of this native salamander in *Eucalyptus* dominated habitats. Our findings prompt further experimental work to determine the mechanisms causing these microbial changes and their potential effect on the fitness of native fauna following invasion.

## ACKNOWLEDGEMENTS

We thank Shannon Buttimer and Natasha Septanova for their help in field collections. We also thank Albert Tang, Carissa Tinoco, and Shannon Buttimer for their help in the laboratory. We are grateful to Andrea Jani and one anonymous reviewer for their comments and suggestions to the text.

### Funding

This work was supported by the National Science Foundation (Nos. 1708926, 1354241, and 1711032). The funders had no role in study design, data collection and analysis, decision to publish, or preparation of the manuscript.

### Grant Disclosures

The following grant information was disclosed by the authors:
National Science Foundation: 1708926, 1354241, and 1711032.

### Competing Interests

The authors declare that they have no competing interests.

### Author Contributions

- Obed Hernández-Gómez conceived and designed the experiments, performed the experiments, analyzed the data, prepared figures and/or tables, authored or reviewed drafts of the paper, and approved the final draft.
- Allison Q. Byrne conceived and designed the experiments, performed the experiments, analyzed the data, prepared figures and/or tables, authored or reviewed drafts of the paper, and approved the final draft.
- Alex R. Gunderson conceived and designed the experiments, performed the experiments, authored or reviewed drafts of the paper, and approved the final draft.
- Thomas S. Jenkinson conceived and designed the experiments, performed the experiments, authored or reviewed drafts of the paper, and approved the final draft.
- Clay F. Noss conceived and designed the experiments, performed the experiments, authored or reviewed drafts of the paper, and approved the final draft.
- Andrew P. Rothstein conceived and designed the experiments, performed the experiments, analyzed the data, authored or reviewed drafts of the paper, and approved the final draft.
- Molly C. Womack conceived and designed the experiments, performed the experiments, analyzed the data, authored or reviewed drafts of the paper, and approved the final draft.
- Erica B. Rosenblum conceived and designed the experiments, authored or reviewed drafts of the paper, and approved the final draft.

### Animal Ethics

The following information was supplied relating to ethical approvals (i.e., approving body and any reference numbers):

We handled all salamanders following a protocol approved by the University of California, Berkeley Animal Care and Use Committee (protocol # AUP-2015-01-7083-1).

## Field Study Permissions

The following information was supplied relating to field study approvals (i.e., approving body and any reference numbers):

Field experiments at Tilden Regional Park and Wildcat Canyon Regional Park were approved by East Bay Regional Parks under permit # 965.

## DNA Deposition

The following information was supplied regarding the deposition of DNA sequences:

Raw 16S rRNA V4 amplicon sequences are available at GenBank Bio Project PRJNA574188 (Soil: SRR10190455–SRR10190486, Skin: SRR10190494–SRR10190543).

## Data Availability

Raw body morphology measurements and R analysis code available in a Supplemental File.

## Supplemental Information

Supplemental information for this article can be found online at http://dx.doi.org/10.7717/peerj.8549#supplemental-information.

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
