# Peer review of "Invasive vegetation affects amphibian skin microbiota and body condition"

_PeerJ, doi:10.7717/peerj.8549_

## Round 0.1 · original submission · Major Revisions

Both reviewers were pleased with this manuscript, commenting that it was straightforward to read and review and complimentary of both the science and writing. Both had a number of minor revisions, and both suggested that you put some significant effort into evaluating the OTUs that differed between treatments. There are a variety of ways to do this, including simple linear models such as t-tests, regressions, anovas, as well as more modern approaches such as lefse (suggested by at least one reviewer) or any of a variety of discriminant analyses (eg indicator species analysis, random forest, etc. as suggested by reviewers.). I look forward to your revised manuscript.

[# Staff Note: the comments from Rev 1 are predominantly in their attached PDF #]

·

Basic reporting

Excellent. See attachment for details.

Experimental design

Excellent. See attachment for details.

Validity of the findings

Excellent. See attachment for details.

Additional comments

This is a well-designed, rigorous study and the manuscript is exceptionally well written and transparent. I had a few questions and suggestions for minor revisions. Please see attached document for details, but in a nutshell, my main comments were:
- I wondered why there were no tests to determine which ASVs differed among treatments.
- I asked for clarification on whether there were differences in salamander microbial community compositional among treatments
- I had a question/suggestion about how you tested the extent of filtering occurring at the host-environment interface.

Reviewer 2 ·

Basic reporting

The authors have presented the main objective of the study, the results and conclusions clearly. The article is very well written and easy to understand.

As a minor comment, I suggest that the results from the Bd analysis be presented in the abstract, particularly because the abstract currently mentions that individuals were tested for Bd. This could be done briefly (e.g. in one sentence that mentions that the incidence of Bd in the study was very low: one Bd+ individual).

Experimental design

Overall, the study was well designed to assess how Eucalyptus invasion might influence amphibian skin bacterial communities and body conditions.

As minor comments:

1- While the introduction clearly explains how the amphibian skin bacterial communities can be a good model system to assess the impact of Eucalyptus invasions on host-associated microbial communities, I believe that the knowledge gap could be better defined here (please see general comments).


2- The statistical analyses conducted in this study are appropriate for assessing the changes across the two type of forests (Eucalyptus vs. Quercus dominated forests) in the skin bacterial communities and body condition of Batrachoseps attenuatus. However, I recommend that the authors include more details about the statistical analyses. For instance, although details are provided in the R-script, it is relevant to mention: 1) that normality tests of response variables were conducted; 2) the error distribution used in the mixed-effect models; and 3) that significance of predictor variables was calculated with likelihood ratio tests. In addition, it would be informative to readers to know which R functions and packages were used for the analyses.

3- In addition, as a suggestion, the presentation of the results from this study could be improved by conducting an analysis showing the differentially abundant ASVs/indicator ASVs across sampling sites, which explain a larger portion of the observed variation in community structure. For visualization purposes, the identified indicator ASVs could be presented either in a heatmap or a stacked-barplot showing the differences in relative abundance of the ASVs (at the phyla level or family level, if possible) across sites or forest types. An indicator species analysis or a linear discriminant analysis Effect Size (LEfSe) might be used.

Validity of the findings

In this study, the authors aimed to determine the potential effects of Eucalyptus dominated forest on the skin-bacterial communities and body conditions of the salamander species Batrachoseps attenuates. This study would be a great contribution to our current understanding on the ecology of amphibian-associated microbial communities, especially since this study focused on a threat to amphibian diversity (i.e. invasive species) that has not yet been explored in the context of skin bacterial communities.

Additional comments

Line 48 – It would be good to be more specific in describing the effect of invasive plants on the ‘patterns’ of native communities. For instance, could it be an effect on the natural diversity pattern of the native communities across a gradient? or a change in the structure of the native communities by changing the relative abundance of dominant species?

Line 68 – Please replace ‘to’ with ‘in’.

Line 72 – Consider replacing ‘Cascading’ with ‘Resulting’.

Lines 86-88 – It seems that this is a good place to highlight the knowledge gap and/or further explain the rationale of the study.

Line 89 – Please clarify the phrase ‘potential for invasive vegetation-induced changes’ to something like ‘…potential changes induced by invasive vegetation on the microbial communities…’.

Line 92 – I would recommend to start this sentence with ‘Specifically,’ since this sentence objectively describes the approach taken to assess the changes induced by the invasive plant on the skin bacterial communities.

Lines 118-120 – It is not totally clear if the same sites from Sax (2002) were used in this study. If so, it would be good to clearly refer to Sax (2002, Table 1) in order to direct the readers to where they can find the elevation, slope, and slope orientation of the study sites.

Line 122 – Please specify the new site in parentheses: (N#).

Line 127 – Were the salamanders caught using gloves? If so, was a new pair of gloves used for each salamander? Please provide such details.

Line 129 – It seems that there might be a discrepancy between the swabbing method described in Hernández-Gómez et al. (2017b) and that used in this study. Hérnandez-Gómez et al. (2017b) mentioned that individuals were swabbed with cotton tip swabs for 30 seconds, whereas it is reported here that individuals were sampled by applying the cotton swab 30 times. Please clarify.

Line 131 – It would be good to mention how the authors controlled for recapturing the same individuals during the field surveys.

Lines 144-146 – Assuming that the original concentration of the master mix is 2X, a 15 ul final volume seems insufficient given the reported concentrations of primers and the volumes of DNA template and sterile water. Thus, I believe that the total reaction volume might not be correct. It seems to me that it should be 25 ul. Please verify.

Lines 152-153 – Same comment as above.

Line 176 – Please add “ 6:e27295v2 https://doi.org/10.7287/peerj.preprints.27295v2” after the words ‘PeerJ Preprint’ in the Boylen et al. (2018) reference.

Line 202 – Perhaps the categories ‘Eucalyptus/Quercus skin’ could alternatively be call ‘Eucalyptus/Quercus salamander-skin’.

Line 204 – I would suggest a minor change in the subtitle: ‘Effects of Eucalyptus invasion on soil bacterial composition, diversity, and stability:’.

Line 210-211 – Considering that the authors are using beta diversity metrics that take into account community composition (unweighted UniFrac) and also relative abundances (weighted UniFrac and Bray-Curtis), I think that the authors could say that they are assessing not only the changes in ASV composition, but also in community structure.

Line 220 – Same comment as above. Suggested title: ‘Effects of Eucalyptus invasion on B. attenuates bacterial composition, diversity, and stability:’

Lines 224-226 – It is not clear in this analysis whether the calculated proportion of shared ASVs at the individual-level, or a presence-absence matrix of all shared ASVs, was used as response variable. The error structure used in the described GLMM (i.e., binomial) suggests that presence-absence data was used. Please clarify.

Line 232 – Same comment as the one from lines 210-211.

Line 251-266 – I recommend to remove the description of the protocol used to detect and quantify Bd from the ‘Statistical Analysis’ section, and to add a new sub-section just for the Bd protocol – probably right before the ‘Statistical Analysis’ section.

Lines 268-273 – I would recommend to add this information in the ‘Material & Methods’ section. Perhaps the number of salamanders sampled could go within the ‘Field Methodology’ sub-section, and the information associated with the sequence processing, including the SRA number, could go within the ‘Microbiota sequence analysis’ sub-section.

Line 277 – Please report the ‘F values’ of the Adonis tests as ‘pseudo-F’, given the permutational approach of the test.

Line 285-290 – I would recommend to report in this text the number of core ASVs associated with the sample types (i.e., Eucalyptus soil, Quercus soil etc.), as well as the number of these core ASVs per phyla.

Line 287 – Are the values in parenthesis an average of the ‘% of reads per sample’ across all samples? Please verify.

Line 293 – It would be informative to provide an average and a standard deviation of the calculated proportion of shared ASVs between the skin of salamanders and their corresponding soil samples across the two habitat types.

Line 311-318 – same comment as the one from lines 285-290.

Line 325 – The prevalence of infection was calculated using 51 as the total number of sampled individuals, however, based on the number of samples swabbed for the microbial analysis, it seems that the total number of samples should be 50. Please verify.

Line 355 – Please replace ‘seems to’ with ‘might’.

Line 360 – It seems that changes in relative abundances of core taxa might be associated with changes in community structure rather than with alpha diversity directly. Perhaps it would be good to mention a potential mechanism by which changes in relative abundance of core taxa might directly influence alpha diversity (e.g., increased probability of a less abundant bacteria to disappear due to competitive-exclusion). Another option could be to just remove the word ‘consequence’.

Line 368-370 – Considering that the authors determined a lower alpha diversity in the skin bacterial communities of the salamanders from the Euclyptus dominated forests, it would be relevant to expand this argument with a brief discussion about what the implications of the observed reduction in alpha diversity might be in terms of the salamanders’ health/fitness.

Line 416 – This sentence states that body condition was not correlated with skin microbiome characteristics, however, no methods neither results testing this statement can be found throughout the manuscript. Please verify.

Figure 3 – Please add (D, E, F) after ‘… and Batrachoseps attenuates skin microbiota samples’.

---

## Round 0.2 · accepted · Accept

Both reviewers indicated that you addressed their (relatively minor) comments thoroughly and we are pleased to advance the article to publication.

Reviewer 2 ·

Basic reporting

No comment.

Experimental design

No comment.

Validity of the findings

No comment.

Additional comments

The comments and suggestions have been adequately addressed.